# Non-Invasive Prenatal Screening: The First Report of Pentasomy X Detected by Plasma Cell-Free DNA and Karyotype Analysis

**DOI:** 10.3390/diagnostics12071591

**Published:** 2022-06-29

**Authors:** Luigia De Falco, Teresa Suero, Giovanni Savarese, Pasquale Savarese, Raffaella Ruggiero, Antonella Di Carlo, Mariasole Bruno, Nadia Petrillo, Monica Ianniello, Ciro Scarpato, Camilla Sarli, Antonio Fico

**Affiliations:** 1AMES-Centro Polidiagnostico Strumentale srl, 80013 Naples, Italy; teresasuero@alice.it (T.S.); giovanni.savarese@centroames.it (G.S.); pasquale.savarese82@gmail.com (P.S.); raffaella.ruggiero@centroames.it (R.R.); dicarloantonella@hotmail.it (A.D.C.); mariasolebruno@libero.it (M.B.); nadia.petrillo@centroames.it (N.P.); monica.ianniello@centroames.it (M.I.); centroames@libero.it (A.F.); 2Fondazione Genetica per la Vita Onlus, 80132 Naples, Italy; 3Responsabile Ambulatorio Medicina Prenatale, P.O.S. Giuliano, 80014 Naples, Italy; dott.ciroscarpato@gmail.com; 4Dipartimento di Medicina Molecolare e Biotecnologie Mediche, Università di Napoli Federico II, 80138 Naples, Italy; camilla.sarli@gmail.com; 5CEINGE Biotecnologie Avanzate, 80131 Naples, Italy

**Keywords:** non-invasive prenatal screening (NIPS), sex chromosome anomaly, prenatal screening, quantitative fluorescent-polymerase chain reaction (QF-PCR), karyotype

## Abstract

Pentasomy X is a sex chromosome anomaly caused by the presence of three extra X chromosomes in females (49,XXXXX instead of 46,XX) and is probably due to a nondisjunction during the meiosis. So far, only five cases prenatally diagnosed were described. The main features in 49,XXXXX karyotype include severe intellectual disability with delayed speech development, short stature, facial dysmorphisms, osseous and articular abnormalities, congenital heart malformations, and skeletal and limb abnormalities. Prenatal diagnosis is often difficult due to the lack of a clear echographic sign like nuchal translucency (NT), and mostly cases were postnatally described. We report the first case of a 49,XXXXX female that was detected by non-invasive prenatal screening (NIPS), quantitative fluorescence polymerase chain reaction (QF-PCR) and a fetal karyotype.

## 1. Introduction

Pentasomy X is a sex chromosome anomaly caused by the presence of three extra X chromosomes in females (49,XXXXX instead of 46,XX) and is probably due to a nondisjunction during the meiosis, either maternal or combined maternal and paternal in origin [1,2]. The true incidence of pentasomy X is unknown but an approximate incidence of 1 in 85,000 was reported comparing to 49,XXXXY syndrome in male live births [3].

The diagnosis is usually made postnatally when specific finding are observed: intellectual disability, variable growth deficiency, Down syndrome-like facial dysmorphy, hypogenitalism and other malformations, especially craniofacial anomalies, cardiovascular and skeletal anomalies [4]. According to a review of the postnatal pentasomy X cases published, intellectual disability and development retardation are the only clinical aspects in common in all the cases reported with a late symptoms onset [4]. Immunoglobulin anomalies, an increased susceptibility to infection and very recently epilepsy and cerebral leukodystrophy have also been reported [5]. Approximately less than 40 cases have so far been reported in the literature [6] with only five pentasomy X cases diagnosed prenatally, because of the absence of indicative echographic patterns for abnormalities caused by the chromosomal deviations which limits the possibility of the prenatal diagnosis [2,7,8,9]. Non-invasive prenatal screening (NIPS), based on cell-free DNA (cffDNA) sequencing analysis, allows the detection of the most common autosomal, including trisomy 21 (T21), trisomy 13 (T13) and trisomy 18 (T18), as well as sex chromosome aneuploidies [7,8,9].

In many countries, NIPS is generally offered as a screening test for T21, T18 and T13 based on evidence of high sensitivity and specificity [10], while more caution on the abnormalities of the sex chromosome is usually adopted, since the occurrence of maternal mosaics and lack of visible features at birth complicate assessment of test performance [11,12]. In addition, a positive NIPT screening for a Sex Chromosome Aneuploidy (SCA), could help parents and health care providers to better plan relevant healthcare interventions and molecular investigations. We report the first case of a 49,XXXXX female that was detected by non-invasive prenatal screening (NIPS), quantitative fluorescence polymerase chain reaction (QF-PCR) and a standard fetal karyotype.

## 2. Materials and Methods

### 2.1. Clinical Report

A 31-year-old pregnant woman primigravida, at 11 + 2 weeks of gestation, without ultrasound evidence and a particular indication or a remarkable family history, came to our center for non-invasive prenatal screening (NIPS). The pregnant woman reported apparent good health, with the exception of hyperprolactinemia due to pituitary microadenoma. Both the pregnant woman and her partner were negative for genetic or chromosomal diseases. They did not report consanguinity. Nuchal translucency (NT) and fetal anatomy were found normal during a routine ultrasound scan. The pregnant woman underwent amniocentesis at 16 weeks, and 16 mL of the amniotic fluid was retrieved for QF-PCR and karyotype to investigate the NIPT results.

### 2.2. cffDNA Isolation from Plasma and NIPS Analysis

For NIPS analysis, about 10 mL of peripheral blood was collected from the pregnant women in Streck blood collection tubes. For plasma isolation, the blood sample was first centrifuged at 1600× *g* for 10 min at 4 °C to separate the plasma from peripheral blood cells. Cell-free DNA from 900 μL of maternal plasma was extracted using the QIAamp DNA Blood MiniKit (Qiagen, Hilden, Germany) following the manufacturer’s protocol. NIPT analysis was performed using the VeriSeq NIPT Solution v2 bioinformatic pipeline (Illumina Inc., San Diego, CA, USA) based on the paired-end sequencing technique. The assay can report the results as Basic, with reporting for common trisomies and sex chromosomes (if selected), and Genome-wide analysis if the detection of the genome-wide fetal anomalies were included (including rare autosomal aneuploidies and partial deletions and duplications ≥  7 Mb) [13,14]. The VeriSeq NIPT Assay Software v2 (www.illumina.com/NIPTsoftware, accessed on 14 December 2021) was used for data analysis of the aneuploidy status and fetal fraction from cffDNA. Sample results were classified using the VeriSeq NIPT Solution v2 Assay Software and analysis of “raw data” as reported previously [14,15].

### 2.3. DNA Extraction, QF-PCR and Karyotype Analysis

Amniotic fluid was drawn and genomic DNA was extracted from the amniocyte using the QIAamp DNA blood Mini Kit (Qiagen, Hilden, Germany). Devyser Compactv3 QF-PCR kit (QF-PCR;Devyser Compactv3, Devyser, Stockholm, Sweden) was used for rapid aneuploidy detection as previously described [16]. The amplified DNA samples were separated by electrophoresis using an ABI 3130xl Genetic Analyzer, and the analysis of each allele for specific markers was performed using the GeneMapper Software ver. 4.0 (Applied Biosystems, Waltham, MA, USA). Two separate tissue culture flasks were used for karyotype analysis. Giemsa Banding (GTG-banded) analysis of amniotic fluid was performed following standard laboratory protocols. A total of 40 metaphases were analyzed with the CytoVision software (CytoVision, AB Imaging).

## 3. Results

### 3.1. NIPS Analysis

NIPS analysis on cffDNA of the pregnant woman did not detect aneuploidy in chromosomes 21, 18 and 13 and the DNA fetal fraction was of 8%. The absence of a Y chromosome was also reported. Moreover, this sample was classified as *Anomaly Detected* by the VeriSeq NIPT Solution v2 assay reporting an XXX result. In addition, the analysis of “raw data” rather than the final report as “Aneuploidy detected” or “No Aneuploid detected” allowed us to note additional X chromosomes copies with respect to the classical 47,XXX. In particular, the observed NCV_X and LLR scores for trisomy X in the “raw data” were higher than expected in the 47,XXX clinical validated cases (internal validated data not shown) and the case was considered a complex sex chromosomal abnormality. During the post-genetic counselling, the family was informed about the NIPS results and were offered an ultrasound scan before the mother’s decision. Ultrasonographic investigations at 20 weeks of gestational age ruled out major fetal malformation (Table 1). Due to the cffDNA results, the pregnant was offered an invasive prenatal diagnosis for additional investigations.
diagnostics-12-01591-t001_Table 1Table 1Review of the literature: clinical features in cases prenatally diagnosed.Case ReportMaternal AgeUltrasound FindingsInvasive Prenatal TestClinical OutcomeClinical Follow-Up FindingsMartini et al., 1993 [7] *39Growth restriction, radioulnar synostosis16 weeks(Second trimester)Amniocentesis(18 weeks)Termination of pregnancyat 20 weeks, autopsyHypertelorism, slight mongoloid slant, radioulnar synostosis, small uterus andhypoplastic ovaries depleted of oocytesMyles et al., 1995 [8]26Dandy–Walker malformation, Hydrocephaly, Ventricular septal defect, Polyhydramnios, Growth restriction(Third trimester)Amniocentesis(33 weeks)Born at 39 weeks with caesareansection, the infant diedat 134 days of ageHypertelorism, Broad flat nasal bridge,small posteriorly rotated ears,short neck, bilateral clinodactyly of fifty digits,ventricular septal defectCheng et al., 2008 [2]29Increased nuchal translucency (3.2 mm), fetal nasalbone absence, Bilateral neck, cysts, Ventricular septal defect(First trimester)Chorionic villoussampling (11 + 5 weeks)Termination of pregnancyat 15 weeks, no autopsyCoarse facial features, low-set ears, depressednasal bridge, generalized edemaAytac et al., 2012 [9]26Increased nuchal fold, Pleural effusion, Subcutaneous edema, Ascites, Bilateral hand clinodactyly (Second trimester)Amniocentesis(17 weeks)Termination of pregnancy, no autopsyNo further information about this casePirollo et al., 2015 [4] *39No major fetal malformation, increased nuchal fold and early, symmetric growth restriction(Second trimester)AmniocentesisTermination of pregnancyat 20 weeks, autopsyAbsence ofsignificant morphological alterationPresent report "31Absence of major fetal malformationAmniocentesis(16 weeks)Termination of pregnancy at 20 weeks, no autopsyAbsence ofsignificant morphological alteration* Advanced maternal age has been the only indication for invasive prenatal test. " Sex Chromosomal Aneuploidy (SCA) detection at non-invasive prenatal screening.

### 3.2. Cytogenetic Analysis

QF-PCR on DNA extracted from the amniotic fluid showed a profile consistent with fetuses disomic for 21, 18 and 13and the presence of more than one X chromosome and the absence of the SRY, confirming the genotype suspected by NIPT (Figure 1). In particular, all informative autosomal STR markers demonstrated a normal 1:1 marker ratio (Figure 1A). The X chromosome counting markers (T1 and T3), which represent autosomal chromosomes compared with chromosomes X, were in the ratio more than 1:3 with respect to X chromosomes, consistent with the dosage of more than 3 X chromosomes (as in Triple X Syndrome). Informative X chromosomal STR markers (X1, X3, X9) demonstrated abnormal marker ratios, and were consistent with the dosage of more than three X chromosomes. In particular, X3 and X9 showed three peaks of different height/area suggesting the presence of three X populations, containing two X each. Informative pseudo autosomal STR markers (XY2 and XY3) also demonstrated an abnormal marker ratio, approximately indicative of the presence of five X chromosomes. Karyotype analysis revealed a 49,XXXXX genotype (Figure 1B). The parents were informed about these findings and, after a genetic counselling, decided for the pregnancy to be terminated in a local hospital in their home town. We were unable to obtain additional information about this case.


## 4. Discussion

Pentasomy X (49,XXXXX) is a very rare aneuploidy involving sex chromosome X. It is characterized by a variable phenotype in females. Pentasomy X was first described in 1963, by Kesaree and Wooley [17], and only around 30 children with a 49,XXXXX karyotype were reported so far [18]. The pathogenesis of pentasomy X is probably caused by successive meiotic non disjunctions, either maternal or combined maternal and paternal in origin [19].

Pentasomy X is characterized by developmental delays, craniofacial anomalies (microcephaly, micrognathia, plagiocephaly, hypertelorism, up-slanting palpebral fissures, a flat nasal bridge and ear malformations), musculoskeletal abnormalities and cardiovascular malformations [4,20]. Pubertal development of affected girls is delayed, and fertility is assumed to be reduced [1,21,22]. Additional findings reported in patients with pentasomy X were immunoglobulin anomalies and an increased susceptibility to infection, thenar atrophy, epilepsy and cerebral leukodystrophy [5,18]. The diagnosis of pentasomy X is usually ascertained postnatally and the prenatal diagnosis of the disease is generally fortuitous. To date only five pentasomy X cases in the literature have previously been diagnosed prenatally [2,4,7,8,9] (Table 1); even more interesting is that detection was made especially in the second/third trimester of pregnancy when major malformations are detectable [4,7,8,9]. In the first case, Martini et al. [9] described a fetus with pentasomy X detected at 18 weeks of gestation with growth restriction and radio-ulnar synostosis determined on ultrasonographic examination [7]. The second case was diagnosed at 33 weeks’ gestation, when ultrasound revealed a Dandy–Walker malformation, hydrocephaly, a ventricular septal defect, hypertelorism and polyhydramnios [8].

The third case was detected at 11 weeks’ gestation by first-trimester nuchal translucency measurement done as part of routine ultrasound screening for Down syndrome [2]. Three-dimensional ultrasound demonstrates increased nuchal translucency thickness, bilateral neck cysts and general edema, and cytogenetic analysis performed by chorionic villus sampling revealed a homogeneous 49,XXXXX karyotype [2].

The fourth pentasomy X case described was discovered because a transient non immune hydrops fetalis occurred at 17 weeks’ gestation in a healthy 26-year-old woman [9]. On ultrasonographic examination, the nuchal translucency was increased and subcutaneous edema, ascites in fetal abdomen and clinodactyly in both hands of fetus was noted [9]. In the fifth case, Pirollo et al. diagnosed a female 49,XXXXX at 20 weeks of gestational age in a 39-year-old woman. She underwent invasive prenatal diagnosis for her advanced maternal age without any other known risk factor [4].

In our report we described the first case of a 49,XXXXX female that was detected by NIPS, QF-PCR and a fetal karyotype in a healthy 31-year-old pregnant woman, at 11 + 2 weeks of gestation. Maternal age as a predisposing factor has been evaluated in previous studies. In one study where 23 cases of pentasomy X syndrome were analyzed, the average maternal age was 27 years [23]. In another study, Pirollo et al., showed that the age was <35 years in 81% of 21 cases of pentasomy X syndrome [4]. These results showed that there is no correlation between advanced maternal age and an increased risk of pentasomy X syndrome. This finding is also in agreement with our case, where the maternal age at conception is 31 years old. The absence of early and clear echographic signs in our case confirmed that the prenatal diagnosis of this syndrome remains a challenge. Hence, the importance of continuing to characterize pentasomy X syndrome cases and to suggest NIPS as routine noninvasive tool screening for pregnant females in order to help in the early diagnosis of this condition. In our case the NIPS analysis reported a sex chromosomes aneuploidy detected as XXX genotype. As previously reported [24], the evaluation of the “raw data” for this sample and the comparison of the NIPT results to all clinical validated Triple X cases ([15] and unpublished data) allowed us to suspect the presence of more than three X chromosomes. Chromosomal analysis on amniocytes from the amniotic fluid showed the 49,XXXXX karyotype, confirming our suspicion.

Although the non-invasive prenatal test was not able to indicate the exact number of X chromosomes copies, the cytogenetic analysis on DNA extracted from the amniotic fluid allowed discrimination between the Triple X Syndrome and the pentasomy X. Trisomy X (47,XXX) syndrome is the most common SCA (up to 1:1000 live births) although it is not always clinically identified [25]. In fact, some individuals are only mildly affected or asymptomatic, and it is estimated that only 10% of individuals with trisomy X are actually diagnosed [26,27]. Moreover prenatal diagnosis rates of 47,XXX are set to increase due to the introduction of NIPS [28].

Since its commercial launch in 2011, NIPS has permitted screening for T21, T18 and T13 with high specificity and sensitivity in both high-and low-risk populations [29,30], while more caution on the abnormalities of the sex chromosomes is usually adopted. cffDNA is derived from both maternal and placental tissues, thus several biological factors such as maternal somatic mosaicism, undiagnosed maternal SCA and maternal copy-number imbalance can influence the accuracy of NIPS [11,12].

Indeed, the diagnosis of SCAs is usually not the primary diagnosis of interest in prenatal genetic testing and, once a SCA is identified, practitioners are faced with genetic counseling regarding the prognosis and treatment of affected fetuses. Knowing that a fetus is affected with an SCA can give the parents a greater awareness of the condition and the clinicians a clear plan to adopt for the healthcare interventions, for instance, fetal echocardiograms to make sure that there are no cardiac abnormalities in fetuses with 45,X (Turner syndrome) [31] or testosterone supplements earlier on after birth, which can improve the health outcome of affected males [32]. Pentasomy X is a rare chromosomal abnormality and only five cases prenatally diagnosed have been described until now. Our case was prenatally diagnosed at 12 weeks of gestational age in a 31-years old woman with NIPT, the results of which suggested the presence of a sex chromosomes aneuploidies. Pentasomy X is a rare condition and usually the availability of little data during the prenatal period make genetic counselling complicated in order to give detailed clinical information to the patient about the disorder. Also, different clinical phenotypes among X chromosome aneuploidies exist, so 47,XXX versus 49,XXXXX can complicate the genetic post-test counselling as well as the clinical outcome decision.

Finally, the case of pentasomy X described in this paper deserves consideration because of rarity of the disease and the importance of the consequences. X polysomy in humans leads to developmental, physical and mental disorders. Therefore, the inclusion of this screening in routine antenatal investigations should be considered, because the absence of a correlation between advanced maternal age and increased risk of pentasomy X and ultrasound signs typical of this syndrome.

The inclusion of a non-invasive test makes it possible to detect cases even in the absence of other risk factors, expanding the clinical coverage of sex chromosome abnormalities.

## 5. Conclusions

Pentasomy X is a sex chromosome anomaly caused by the presence of three extra X chromosomes in females (49,XXXXX instead of 46,XX) and is probably due to a nondisjunction during the meiosis. Prenatal diagnosis is often difficult due to the lack of indicative ultrasonographic findings and the rarity of described cases. Although the clinical experience with non-invasive prenatal screening (NIPS) for fetal sex chromosomes using sequencing of maternal plasma cell-free DNA is limited, the presented case demonstrates the clinical utility of cffDNA based screening in the detection of sex chromosome aneuploidy in order to improve the prenatal genetic counselling, and giving detailed clinical information to the patient about the disorder.

## Figures and Tables

**Figure 1 diagnostics-12-01591-f001:**
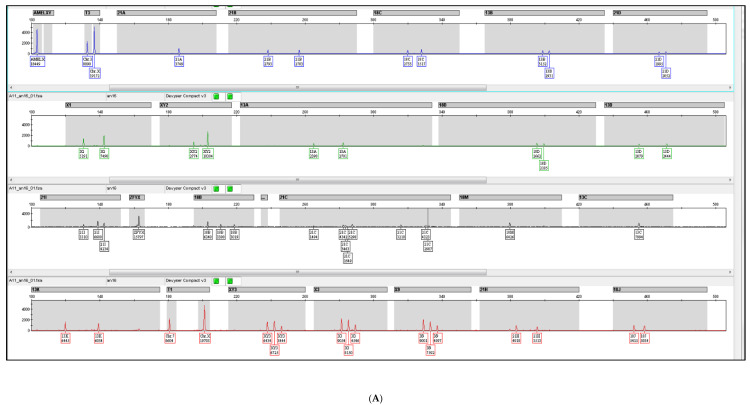
Cytogenetics characterization of amniotic fluid. (**A**) QF-PCR analysis. Informative STR markers on all autosomal chromosomes demonstrate a normal 1:1 marker ratio. The presence of AMELX and the absence of AMELY and SRY is consistent with female gender. The T1 and T3 markers are non-polymorphic (non-STR) X chromosome counting markers that may be used to determine the number of X chromosomes. The abnormal marker ratio (1:5) of the X chromosome counting markers (T1 and T3) is consistent with an abnormal female X chromosome dosage. Informative X chromosomal STR markers (X1, X3, X9) demonstrate abnormal marker ratios, and is consistent with the dosage of more than three X chromosomes. Informative pseudoautosomal STR markers (XY2 and XY3) demonstrate an abnormal (4:1) marker ratio; (**B**) GTG banding analysis of amniotic fluid show a 49,XXXXX karyotype.

## Data Availability

All data generated or analyzed during this study are included in this published article. Protocols and deidentified, aggregated data that underlie the results reported in this article are available for non-commercial scientific purposes upon reasonable request from the corresponding author. For privacy reasons raw data are not publicly available.

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
