# Peer review of "Non-Invasive Prenatal Screening: The First Report of Pentasomy X Detected by Plasma Cell-Free DNA and Karyotype Analysis"

_diagnostics, 2022, doi:10.3390/diagnostics12071591_

Round 1

Reviewer 1 Report

This is a very interesting case, the precise genetic diagnosis allowed for proper prognostic evaluation. It deserves publication after minor revision.

Suggestions are in order:

- Diagnostic methods are appropriate and well described. However the number of analyzed methaphases during amniocentesis should be specified.

- Discussion should be significantly shortened

Author Response

Reviewer 1

  • Diagnostic methods are appropriate and well described. However, the number of analyzedmethaphases during amniocentesis should be specified.

  • As suggested by the reviewer we added the number of the analysed methaphases

  • Discussion should be significantly shortened

As suggested by the reviewer we have reduced the length of the discussion

Reviewer 2 Report

Non-invasive prenatal screening: the first case of Pentasomy X based on plasma cell-free DNA and karyotype analysis

The manuscript reported the first Pentasomy X case which screened positive by non-invasive prenatal screening cell free DNA. The cell free DNA screening test showed abnormal dosage of X chromosome. Later, the diagnosis with 49,xxxxx was confirmed by QF-PCR and karyotype.

The authors emphasized the benefits of using this cell free DNA screening technology in routine prenatal care since patients may have normal prenatal ultrasound and there is no risk factor which may help to identify the risk to have a baby with sex chromosome aneuploidy. The screening may not accurately report the copy-number of sex chromosome therefore a confirmatory test is necessary.

Comments

1.       The title can be misinterpreted as this is the first case of pentasomy x diagnosed by karyotype. This is the first report of pentasomy x case first identified by NIPS

2.       Please check ACMG recommendation, the cell free DNA screening test should be called non-invasive prenatal screening or NIPS not NIPT since this is a screening test not a diagnostic test.

3.       Page 1 line 36, please clarify “all cases reported with a late onset”, should it be the first presentation is detected later in life. Pentasomy X has onset at birth, just the first symptoms are first noticed later.

4.       Discussion, line 155-171, the information is similar to what already mentioned in introduction.

5.       Are there any circumstances that may cause false positive or false negative SCA?

6.       Figure, I only saw QF-PCR and karyotype but not NIPS which is actually be the focus of this manuscript.

Author Response

Reviewer 2

- The title can be misinterpreted as this is the first case of pentasomy x diagnosed by karyotype. This is the first report of pentasomy x case first identified by NIPS

As suggested by the reviewer we rewrite the title according the manuscript.

  • Please check ACMG recommendation, the cell free DNA screening test should be called non-invasive prenatal screening or NIPS not NIPT since this is a screening test not a diagnostic test.

As suggested by the reviewer we rewrite the NIPT as NIPS, non-invasive prenatal screening

  • Page 1 line 36, please clarify “all cases reported with a late onset”, should it be the first presentation is detected later in life. Pentasomy X has onset at birth, just the first symptoms are first noticed later.

As suggested by the reviewer we added a sentence which clarify “with late onset”.

  • Discussion, line 155-171, the information is similar to what already mentioned in introduction.

In this section of the discussion we described the previously reported prenatal case to underline that all cases are diagnosed later during the pregnancy (and not for a specific ultrasound findings) and therefore the importance of our case.

  • Are there any circumstances that may cause false positive or false negative SCA?

YES, we added a sentence about this information.

Lane223-225: “cffDNA is derived from both maternal and placental tissues, thus several biological factors such as maternal somatic mosaicism, undiagnosed maternal SCA and maternal copy-number imbalance can influence the accuracy of NIPS[11,12].”

  • Figure, I only saw QF-PCR and karyotype but not NIPS which is actually be the focus of this manuscript.

We have not shown any figure about NIPS, because currently nips results are showed as a report“aneuploidy detected” or “not detected” as reported in methods and result section.In addition, we write in result section:“the analysis of “raw data” rather than the final report as “Aneuploidy detected” or “No Aneuploid detected” allowed us additional X chromosomes copies respect to classical 47,XXX. In particular, the observed NCV_X and LLR scores for trisomy X in the “raw data” were higher than expected in the 47,XXX clinical validated cases (internal validated data not shown) and was considered a complex sex chromosomal abnormality.”

Reviewer 3 Report

I have only three minor comments.

1. I wonder if large Figure 1A can be improved for visibility. In particular, I could not find an X2 (X chromosome STR) maker on the map.

2. The abbreviation "SCA" first appears on line 49. It is here that SCA should be defined as sex chromosome aneuploidy rather than later in line 210.

3. I suggest that "mostly"", "because" and " respect to" on several places (line 19, 40, 47, 98, 113, 244) are in improper use.

Author Response

Reviewer 3

  • I wonder if large Figure 1A can be improved for visibility. In particular, I could not find an X2 (X chromosome STR) maker on the map.

As suggested by the reviewer we improved the Figure 1A for visibility. We did not include X2 STR , because we had an overlap of signals for this mix. We removed the X2 from the text and from the figure.

  • The abbreviation "SCA" first appears on line 49. It is here that SCA should be defined as sex chromosome aneuploidy rather than later in line 210.

As suggested by the reviewer we added the “sex chromosome aneuploidy “on line 49, as first it appears in the text

  • I suggest that "mostly"", "because" and " respect to" on several places (line 19, 40, 47, 98, 113, 244) are in improper use.

As suggested by the reviewer We removed these.